# A Panoramic Localizer Based on Coarse-to-Fine Descriptors for Navigation Assistance

**DOI:** 10.3390/s20154177

**Published:** 2020-07-27

**Authors:** Yicheng Fang, Kailun Yang, Ruiqi Cheng, Lei Sun, Kaiwei Wang

**Affiliations:** 1State Key Laboratory of Modern Optical Instrumentation, Zhejiang University, Hangzhou 310027, China; fangyicheng@zju.edu.cn (Y.F.); rickycheng@zju.edu.cn (R.C.); leo_sun@zju.edu.cn (L.S.); 2Institute for Anthropomatics and Robotics, Karlsruhe Institute of Technology, 76131 Karlsruhe, Germany; Kailun.Yang@kit.edu; 3National Engineering Research Center of Optical Instrumentation, Zhejiang University, Hangzhou 310058, China

**Keywords:** visual place recognition, coarse-to-fine descriptors, panoramas, navigation assistance

## Abstract

Visual Place Recognition (VPR) addresses visual instance retrieval tasks against discrepant scenes and gives precise localization. During a traverse, the captured images (query images) would be traced back to the already existing positions in the database images, rendering vehicles or pedestrian navigation devices distinguish ambient environments. Unfortunately, diverse appearance variations can bring about huge challenges for VPR, such as illumination changing, viewpoint varying, seasonal cycling, disparate traverses (forward and backward), and so on. In addition, the majority of current VPR algorithms are designed for forward-facing images, which can only provide with narrow Field of View (FoV) and come with severe viewpoint influences. In this paper, we propose a panoramic localizer, which is based on coarse-to-fine descriptors, leveraging panoramas for omnidirectional perception and sufficient FoV up to 360${}^{\circ }$. We adopt NetVLAD descriptors in the coarse matching in a panorama-to-panorama way, for their robust performances in distinguishing different appearances, utilizing Geodesc keypoint descriptors in the fine stage in the meantime, for their capacity of detecting detailed information, formatting powerful coarse-to-fine descriptors. A comprehensive set of experiments is conducted on several datasets including both public benchmarks and our real-world campus scenes. Our system is proved to be with high recall and strong generalization capacity across various appearances. The proposed panoramic localizer can be integrated into mobile navigation devices, available for a variety of localization application scenarios.

## 1. Introduction

Nowadays, robotics, unmanned systems, autonomous vehicles, and navigation devices for visually impaired people (VIP) all have precise localization appeals [1,2]. Large numbers of mature positioning technologies have been thoroughly researched in the field of localization and applied to the navigation equipment [3,4,5,6] The rapid development of smartphones and the coverage of mobile Internet in large areas facilitate GNSS-based positioning [7] methods to widely proliferate in mobile phone apps. We can therefore acquire real-time localization information accessibly and conveniently. However, in some GNSS-denied areas, such as multi-story buildings, parking lots, and remote areas with weak satellite signals, GNSS may lose efficacy. Pseudo-satellites [8], also known as “ground satellites”, emit GNSS-like navigation signals from a specific point on the ground, can improve the reliability and anti-jamming ability, thus making localization available in some GNSS-denied areas. WIFI [9] and UWB Ranging [10] are two of the wireless indoor positioning technologies. WIFI is performed by acquiring the Received Signal Strength Indication (RSSI) of mobile devices to surrounding Access Points (APs) to estimate the location, which possesses the advantage of low cost and is appropriate for some meter-level positioning accuracy-needed scenarios. Nevertheless, it heavily relies on the coverage of WIFI signal and is limited in small-scale situations. UWB Ranging technology receives and transmits the extremely narrow pulse in a nanosecond or less to transfer the data, with a bandwidth of GHz magnitude. It shows superior positioning accuracy ranging from 1 cm to 10 cm and perfect real-time performance. Unfortunately, UWB Ranging is subject to interference and is not suitable for crowded places. Therefore, it is more suitable for some specific applications, such as intelligent anchor and robot motion tracking.

The localization approaches mentioned above all have some limitations in specific scenarios indoors or outdoors. While tackling “which point you are” problems, they ignore “what you can perceive in the surroundings”. The capacity to recognize and distinguish environments is as significant as its exact location points in the map during navigation. Sensor fusion localization [11] takes complementary advantages of multi-sensors to provide a solution tackling the above problems to some extent, but the sophisticated fusion scheme and high cost of multiple sensors could be prohibitive in some application scenarios. Fortunately, Visual Place Recognition (VPR) facilitates addressing these difficulties in a purely vision-based measure, which refers to using image retrieval and similarity comparison between query images and database images to determine the best localization result and recognize the surrounding scenes in the meantime [1,2,12,13]. Not restricted by weak signals of satellite or RSSI from APs, VPR is available for extensive application scenarios both indoors and outdoors with only a low-cost and simple camera. In addition, it is possible to effectively tackle both “which point you are” and “what you can perceive in the surroundings” issues. Efficient and robust VPR algorithms can be integrated on navigation devices with mobile computer platforms [14,15,16,17]. VPR may perfectly alleviate the potential hazards caused by inaccurate localization, providing navigation assistance for high-level applications like self-driving cars [2] and guidance of VIP [1].

However, there are so many factors that pose significant challenges to the robustness of VPR algorithms, involving illumination changing, seasonal cycling, viewpoint variations, and so on. Serious appearance variations have been thoroughly investigated [18,19,20]. Large numbers of CNN-based methods in VPR have been proposed and proved successful, either acting on global descriptors to obtain superior appearance-robust, like [21], or paying attention to saliency-focused local landmark based approaches, like [17,22]. However, global descriptors based systems may easily gain perspective sensibility due to the camera bias, and using only local landmarks to locate, it is possible to lose control of the overall environments, due to the potential loss of significant content in the images. Garg et al. [23] have conducted some attempts to implement an appearance-invariant VPR system against totally opposite viewpoints by using only a limited Field of View (FoV) camera. They leveraged visual semantic information and constructed a novel Local Semantic Tensor (LoST) descriptor by using the convolutional feature maps from a state-of-the-art dense semantic segmentation network, and then developed a novel approach for mining semantically salient keypoint correspondences to verify the spatial semantic arrangement of the top matching candidates. However, deviation of camera orientation may easily cause missing of valid image information when not pointing straightly ahead or straightly behind. According to the state-of-the-art works, two points are summarized that matter most in VPR retrieval tasks. One is that the camera’s narrow FoV may greatly restrict the accuracy of positioning and influence the recall possibility, while the other one is that one-stage (global-only or local-landmark-only) based system may not be stable and feasible in face of most challenges.

To cope with the first point, we make our efforts to extend FoV to 360${}^{\circ }$ to alleviate the impacts brought by the limited imaging angle, through introducing panoramas with omnidirectional observations into our localizer. Detailed advantages of introducing panoramas into VPR include:

Both pedestrians and vehicles need to be aware of the environment from entire surroundings. Navigation systems embedded in mobile devices are typically designed to work with a forward-view camera, such as a pinhole camera, or an array of cameras to obtain surrounding views. As the mobile device moves forward, blind spots will appear in one position with the captured images, where the lack of such information may lead to localization failures. For example, the database images stored at this location and the query images captured in the the same location may be taken in completely different directions. Nevertheless, panoramas can provide omnidirectional perception, and capture large-FoV scenes, which can significantly promote the localization reliability.The variations of viewpoints bring severe challenges to VPR. During different traversals, it is common to undergo orientation changes, slightly or drastically. Forward-facing cameras, which usually keep the same directions with mobile devices, feature large variations of lateral displacement and orientation. However, the panoramas cover $360^{\circ }$ omnidirectional information of a location. Under the condition when the vehicles move or the cameras shake, the panoramic information of this position will all be taken in the full-view image. In this way, the panoramas minimize the impact of the viewpoint changing in VPR.

However, panoramas-based VPR algorithms remain challenging and unsolved despite their great advantages. Three dominating reasons are possible to make them intractable. Firstly, it is owing to the scarcity of public large-scale panoramic datasets with ground truth on a continuous route. The requirement of record for double or more times under different appearances is also critical for developing and evaluating VPR algorithms. Secondly, due to the high resolution of panoramas, the efficiency and memory consumption should be carefully considered, as they often entail heavy and time-consuming computation. Thirdly, when panoramas are fed into a forward-facing images trained Convolutional Neural Network (CNN), a great precision downgrade would emerge, due to the huge distinction between conventional and panoramic images. Luckily, attempts on panoramic research never stops, where many state-of-the-art technologies contribute a lot to solving these troublesome issues. [24,25,26] In our panoramic localizer, a panoramic annular lens (PAL) [27] is utilized to relieve these problems in some way, which will be interpreted in detail in the next section.

To tackle the second point, we put forward a two-step coarse-to-fine solution to achieve targeted positioning at each stage, which processes from the global image to local regions. Both global and local descriptors have been used in VPR. Global descriptors are usually extracted from the whole image to represent the holistic scene [20,28]. Local features, such as keypoint descriptors, which can utilize only local statistics, are usually aggregated together to represent the whole image [29,30]. Global descriptors are regarded as robust to global environmental changing, among which CNN descriptors perform competently in VPR tasks and achieve high-level localization precision compared to ordinary handcrafted global descriptors [22,31,32,33,34]. Particularly, NetVLAD [35] descriptors, combining common CNN backbone with a NetVLAD layer, can effectively improve the capacity of recognizing images from distinguishing location and retrieve the best matched location. Nevertheless, only global descriptors are not accurate enough to yield finest localization, although they can perfectly give results in a rough range. Then, local descriptors, which are preponderant in providing detailed information, can determine much finer localization, but at a cost of time exhaustion. Geodesc [36] keypoint descriptors, compared to SIFT, SURF, and ORB descriptors, have a superior keypoint feature learning ability but are hardly used in VPR field. Thus, the combination of global and local descriptors, which forms coarse-to-fine descriptors, has been taken into consideration in our panoramic localizer.

In this paper, a novel panoramic localizer based on coarse-to-fine descriptors, which can be deployed to mobile devices and provide navigation assistance is proposed, in order to solve the problems of appearance variations that VPR is facing, to better perceive surroundings and to determine much more accurate localization. Figure 1 shows the overview of the proposed coarse-to-fine localizer. The contributions of the panoramic localizer reside in the following aspects:The proposed outstanding panoramic VPR algorithm can be divided into two processes: the coarse stage and the fine stage, forming coarse-to-fine descriptors, aimed to cope with appearance changing problems and obtain omnidirectional perception.In the coarse stage, NetVLAD descriptors are implemented in a panorama-to-panorama way from holistic panoramas, instead of training the NetVLAD network by using only forward-facing images. This operation is critical to keep the feature consistency of input and output panoramas. The coarse stage provides rough localization results for finer matching in the subsequent fine stage.In the fine stage, keypoint descriptors are leveraged to detect detailed information, on the condition that the panoramic image planarity can be guaranteed. As deep-learning based keypoint descriptors, Geodesc descriptors have demonstrated a better capacity of capturing critical keypoints and good matches than handcrafted conventional keypoint features [36]. The fine stage gives much finer localization results.A single-shot camera, the panoramic annular lens (PAL), is utilized in our panoramic localizer. We collected a variety of panoramic datasets and made them publicly available to the community, including stitched panoramas from public data for our coarse-to-fine processing, and real-world scenarios collected with a PAL for evaluating the generalizability of our approach. The Chengyuan dataset can be obtained at https://github.com/dachengzihhh/Chengyuan-dataset.The proposed panoramic localizer can be easily deployed on wearable systems for VIP and instrumented vehicles, performing practical navigation tasks when applied in our daily life, as shown in Figure 2.

## 2. Related Work

### 2.1. Panoramic Annular Lens (PAL)

The PAL system is one of the specific single-sense gaze imaging systems [37,38,39]. The basic structure and imaging optical path of PAL are shown in Figure 3. The PAL head unit possesses a much larger caliber than the subsequent relay lenses, which guarantees that wide field-of-view light from the objects (such as a and b, which represent the maximum incident angle) can be shot into the system. Once refraction and twice reflections on PAL head unit can effectively reduce the angle of light shot into relay lenses, to a final image on a relatively small image surface. The imaging range of the PAL system is generally horizontal $360^{\circ }$, at each angle, light a is with the minimum vertical angle that can be received by the PAL system, while light b is with the maximum vertical angle that can be received. Lights a and b and the incident light between them jointly determine the FoV of PAL imaging. PAL features a simpler structure and lower cost, whose size can be adapted to different applications flexibly, and there have been some attempts and research in both industry and academia. Hu et al. [39] proposed a multiple-pinhole rectification of raw PAL images, which allows the undistorted images after multiple-pinhole rectification to satisfy the perspective projection rules, by mapping a PAL image onto an unit sphere, extending the unit sphere to a cube, and then using four virtual pinhole cameras to capture the side surfaces of the cube. Their work provides an excellent panoramic annular processing method to obtain omnidirectional multiple-pinhole plane image groups with only one PAL, which we call panoramic plane images in this article. The annular images taken by PAL avoid the upper and lower part of spherical images, and only leave the intermediate part of images that contain important visual cues for VPR.

### 2.2. Multi-Step Solutions on VPR

Plenty of efforts have been made to confirm that the multi-step localizer outperforms the single-stage based approach in terms of not only the positioning accuracy, but also the robustness. Our previous work has already demonstrated the effectiveness of multi-stage strategies, which process from a global image, and then to a local region, lastly to keypoints [12]. Global features based on GoogleNet pretrained on the Places dataset [40] helped recognize overall environments, while the salient region only kept tactics that acted on critical parts of images, and keypoint descriptors extremely focused on the detailed content. The procedures are cumbersome but gradually reduce the range of candidates, resulting in greatly improved accuracy. Li et al. [25] also presented a coarse-to-fine positioning method, which fully leverages the topometric structure of the street view to conduct a topological place recognition in the coarse stage, while carrying out a metric pose estimation by local bundle adjustment in the fine stage. Their evaluation on a 3 km urban environment trajectory across from viewpoint changes, illumination, and occlusion successfully proves the localization consistency from coarse to fine. Aiming to tackle the issue of large-scale localization applications computing efficiency, Liao et al. [41] proposed a semantic compact map based coarse-to-fine visual localization solution. The authors believed pole-like objects, such as traffic light, lamp, and tree trunks, are reliable landmarks for localization, which are directly extracted from semantically segmented images in the coarse stage. In addition, they utilized a pose alignment module to adjust the pose to a finer position.

### 2.3. Attempts on Panoramic VPR

In the field of panoramic visual positioning, scholars have done extensive and insightful research by deriving from existing pin-hole images based VPR. They aimed to overcome problems of memory usage, computation complexity, and the shortage of large-scale panoramic image dataset. Holliday et al. [42] constructed a scale-robust localization system in robotic mapping applications, by combining deep-learning based object features and SIFT descriptors, which yields improved robustness to scale change. They proposed a non-semantic objects identifying and describing technique. Thanks to the high degree of abstraction of intermediate layer of CNN, it allows these non-semantic objects to be associated between scenes, which can guide the matching of SIFT descriptors, resulting in the scale changing robustness. Cheng et al. [2] also presented a panoramic annular images based method, the Panoramic Annular Localizer, which discarded previous public dataset pretrained passive deep descriptors, but utilized more adaptive active NetVLAD descriptors for the application on scene images. However, the network was trained with common pin-hole images but tested and implemented on large-scale panoramas, which inevitably degrades the localization accuracy to a large extent. The work of Iscen et al. [43] has provided us with insights. They put through a panorama-to-panorama VPR method based on NetVLAD descriptors, by testing through implicit construction of a panorama in the descriptor space, and explicit construction of a panorama in the image space respectively on the street view imagery. It came out that a single panoramic NetVLAD descriptor is preferable to aggregating individual views into a vector in most of the situations for visual localization tasks. In other words, the panorama-to-panorama model was better suited to VPR tasks than other image-to-panorama matching and panorama-to-panorama matching. In the coarse stage of our panoramic localizer, we refer to the similar panoramic-to-panoramic solution by using explicit panoramas to train the NetVLAD network, and then feeding a holistic panorama into the network to obtain a single robust descriptor for one panorama towards accurate VPR.

## 3. Panoramic Localizer: The Proposed Visual Localization System

The specific implementation steps of the panoramic localizer are elaborated in this section. In order to realize accurate localization across severe environmental variations, the positioning is conducted from the coarse stage to the fine stage, and each stage is equipped with robust descriptors to ensure the localization accuracy. The coarse stage can be regarded as the foreshadowing of the fine stage, which finds the rough range of correct localization results for fine matching and greatly reduces the search range for the fine stage. Meanwhile, as the keypoint descriptors utilized in the fine stage are very time-consuming and resource-demanding, once the search scale is reduced, the matching efficiency will be significantly improved.

Therefore, three following successive steps are involved to realize the proposed panoramic localizer. Firstly, we will describe the acquisition of panorama resources. Both the coarse stage and the fine stage have specific requirements on the form of the panoramic images. More precisely, the coarse stage uses the panoramic surround images, while the fine stage leverages the panoramic plane images. Luckily, the panoramic annular images captured from PAL can be processed into two forms, panoramic surround images and panoramic plane images, respectively. Secondly, the coarse stage is based on NetVLAD descriptors from different CNN backbone models. It is constituted of panorama-to-panorama matching, where finally the rough range is chosen by calculating the Euclidean distance between NetVLAD descriptors. Thirdly, the fine stage utilized keypoint descriptors, such as ORB, SIFT, and particularly the deep-learning based geodesc descriptors. The fine stage only searches in the coarse range provided by the last stage, and uses the method of plane mapping (Fundamental matrix) [44] to select the final most accurate localization.

### 3.1. Acquisition of Panorama Resources with PAL

The coarse localization stage needs panoramic surround images to train and perform in the NetVLAD network to satisfy the panorama-to-panorama matching, while the plane mapping (such as the Fundamental matrix based matching) in the fine stage is based on the premise of image planarity, where panoramic plane images are needed. In addition, the transformation of panoramic surround images and panoramic plane images from panoramic annular images imaged by PAL should be easy to get and quick to acquire. Compared to the traditional method that stitches pin-hole images taken from different perspectives into a panorama, panoramic annular images can be imaged once by PAL, which weakens the image quality effect of stitching to a great extent. In addition, panoramic annular images can be processed to panoramic surround images and panoramic plane images through simple programs. In addition, the panoramic plane image processing can effectively eliminate the distortion appeared in the panoramic annular images in the meantime.

The PAL system is different from the image projection relation of pinhole camera due to its wide FoV, so rectification operations are needed. For imaging systems, the mapping relationship between camera coordinate system and pixel coordinate system can be obtained by calibrating internal parameters, and their projection relationship can be described by the virtual unified spherical model, with the camera as the origin. Each point of pixel coordinates corresponds to a vector whose center points to the virtual unit sphere. The cylinder distortion-free rectification of the panoramic view is suitable for human eye observation, for the upper and lower bounds of the visual field remaining flat when the human eye looks around, so projecting the virtual unit spherical onto a cylinder and unfolding the cylinder can obtain the common and familiar panoramic surround image, shown as the upper method in Figure 4c. Nevertheless, the fine stage requires the image planarity, which means that, to conduct the multiple-pinhole rectification operation mentioned in Section 2.1 [39], projecting the virtual unit spherical onto a cube is satisfied, which possesses the same origin as the virtual unit spherical and unfolding of the cube. In this way, we obtain four plane images containing $360^{\circ }$ surrounding information, shown as the bottom method in Figure 4c. Both panoramic surround images and panoramic plane images can be acquired in real time, which theoretically can be easily obtained once the PAL camera calibration results are available. However, even if a PAL is unavailable, the panoramic surround images can be obtained by image stitching or even mobile phone scanning panoramic image function, and the panoramic plane images can be acquired by multi-pinhole cameras, in order to fit our panoramic localizer system.

### 3.2. The Coarse Localization Stage

The coarse stage, selecting the rough range of correct localization, serves as the prior preparation for the fine stage. It makes use of the robust NetVLAD descriptors, in a panorama-to-panorama matching way.

#### 3.2.1. Backbone

A backbone network is the prerequisite to an integrated NetVLAD network, which is usually chosen from prevailing classification networks including AlexNet [45], VGG [46], ResNet [47], MobileNet [48,49,50], and so on. It is worth mentioning that the backbone should discard the last fully connected layers and pooling layers of classification networks to guarantee the output of backbone a fully convolutional structure, to better serve for the input of following NetVLAD layer. The below four classification networks are our primary attempts aimed to determine one with good comprehensive performance.

AlexNet [45] opens the vista of the rapid development of CNNs in the computer vision community, with the simplest structure of the four networks. Evolved from AlexNet, VGG16 [46] overlays the convolutional layer and pooling layer to increase the network depth. ResNet [47] attempts to deepen the network to obtain more complicated features, while finding it fail in training the model in reality due to the vanishing gradient or explosive gradient, so a short-cut unit is added to deploy residual learning. MobileNet [48], based on deep separable convolutions, is a kind of lightweight network designed for mobile and embedded devices, which utilizes deep separable convolutions to compute the cross-channel correlations and spatial correlations, processing from depth-wise convolutions to point-wise convolutions, respectively. MobileNet V2 [49] also adopts a short-cut structure to improve network performance and compress the network scale. Assuming the size of the filters is *K* × *K*, the number of input channels is $C_{in}$, and the number of output channel is $C_{out}$, so the number of floating-point multiplication-adds (FLOPs) of common convolutions can be represented as follows:(1)$$FLOPs_{common}=C_{in}\times K\times K\times C_{out}$$

The FLOPs under deep separable convolutions can be represented as:(2)$$FLOPs_{deep\;separable}=FLOPs_{depthwise}+FLOPs_{point-wise}=C_{in}\times K\times K\times 1+C_{in}\times 1\times 1\times C_{out}$$

Thus, the number of parameters can be reduced by
(3)$$\frac{FLOPs_{common}}{FLOPs_{deep\;separable}}=\frac{C_{in}\times K\times K\times C_{out}}{C_{in}\times K\times K\times 1+C_{in}\times 1\times 1\times C_{out}}=\frac{1}{C_{out}}+\frac{1}{K\times K}∼\frac{1}{K\times K}$$

When adopting deep separable convolutions compared to common convolutions, usually $C_{out}$ is much larger than *K*, so the computation complexity can be cut to nearly 1/K × *K* of that with standard convolutions, which significantly improves the efficiency. The architecture of these four backbones is shown in Table 1.

#### 3.2.2. NetVLAD Descriptors

The NetVLAD module is a learnable feature pooling network, which realizes robust description of input images through weakly-supervised learning on the location recognition dataset. NetVLAD can extract the descriptors with the ability of location determination from the lower level feature map. A standard CNN followed by a NetVLAD layer constitutes the NetVLAD network architecture. We feed panoramic surround images into the NetVLAD network to train the network and to validate the panoramic localizer system, which can be seen as panorama-to-panorama matching. The output of the last layer of standard CNN can be seen as a feature extractor. Then, the NetVLAD layer pools each extracted feature into a fixed image presentation, where the corresponding parameters can be learned in a backward propagating way, to continually optimize the image presentation. In this way, we can finally obtain a NetVLAD descriptor to express the holistic image. In addition, triplet loss function was designed for NetVLAD network to impel the query image to better distinguish the geographically far-away negatives and nearby positives, by providing every piece of training data with a position coordinate. In this way, the image expression ability of the same category will be effectively improved, and the ability of classification from different scenes will be greatly reinforced.

#### 3.2.3. Global Panorama Matching

Each NetVLAD descriptor can represent an input image, but how to compute the similarity between the query images and the database images to find out the rough range remains unsolved. Therefore, we implement Brute Force (BF) [51] into our panoramic localizer, taking into account of the correlation between the images and conducting one-by-one search. We compute the Euclidean distance [52] between query images and database images as follows:(4)$$\begin{array}{cc} d(x,y)= & \sqrt{{\left(x_{1}-y_{1}\right)}^{2}+{\left(x_{2}-y_{2}\right)}^{2}+\cdots +{\left(x_{n}-y_{n}\right)}^{2}}\\ \; & =\sqrt{\sum_{i=1}^{n}{\left(x_{i}-y_{i}\right)}^{2}} \end{array}$$
where
(5)$$x=(x_{1},x_{2},\cdots ,x_{n})$$
represents the feature extracted from query image;
(6)$$y=(y_{1},y_{2},\cdots ,y_{n})$$
represents the feature extracted from database image; and d(x,y) represents the Euclidean distance between the query image and the database image. Obviously, the lower the Euclidean distance is, the more similar the query image and the database image are, which can be the rough range selection criteria.

### 3.3. The Fine Stage

The fine stage conducts one-by-one searching within the coarse range, or in other words the selected top k candidates selected from the coarse stage, and determines finer localization results. In this period, robust keypoint descriptors are performed in our panoramic localizer, to guarantee higher localization accuracy compared to only conducting the coarse stage.

#### 3.3.1. Keypoint Descriptors

There are both handcrafted traditional keypoint descriptors and deep-learning based keypoint descriptors. SIFT is one kind of the handcrafted local features, which remains invariable to rotation, scale scaling, and brightness changing, and shows a stability to a certain extent to angle changing, affine transformation as well as noise. ORB is a kind of BRIEF based fast binary feature vector descriptor, which has the characteristics of rotation invariance and noise suppression. ORB is not as robust as SIFT but possesses a fast detection speed of an order of magnitude higher than SIFT and is often utilized as a real-time keypoint feature detection tool.

Traditional handcrafted keypoint descriptors like ORB, SIFT and so on have been widely studied in the VPR community, while some deep-learning based keypoint descriptors are rarely utilized in localization tasks, although they have much higher adaptation to image contents. Geodesc descriptors, as state-of-the-art deep-learning based keypoint descriptors, offer a novel batch constructed method that simulates the pixel-wise matching and effectively samples useful data for the learning process. The learning procedures are as follows: Firstly, the SIFT keypoints are detected throughout the image (without extraction of SIFT descriptors). Then, these keypoints are cropped as 32 × 32 sized squares. Finally, these 32 × 32 sized squares, which can represent the keypoints, are input into the L2-Net pretraiend on the Hpatch dataset [53] and each output vector can describe a keypoint. The comparison of matching results among ORB descriptors, SIFT descriptors, and Geodesc descriptors is shown in Figure 5.

#### 3.3.2. Keypoint Descriptor Matching

After the extraction, effective measures should be conducted to utilize the keypoint descriptors to determine the best locations. A Fundamental Matrix [44] based on plane mapping is taken into consideration, aimed at finding the mapping relations between keypoints from query images and database images. Fundamental Matrix offers constraints on three-dimensional points to two-dimensional epipolar lines. As shown in Figure 6, there are two images *P*, $P^{\prime }$ taken from different viewpoints. If a random point *x* exists in the left image, and the two images follow the principle of the Fundamental Matrix, the point *x* must be a counterpart to an epipolar line l′ in the second image, and point $x_{1}\prime$$x_{2}\prime$, $x_{3}\prime$ on the l′ is the projection of $x_{1}$, $x_{2}$, $x_{3}$ on the extension line of Ox (O is the camera center of the left image). The Fundamental Matrix can be represented as follows:(7)$$q_{1}F^{T}q_{2}=0$$
where $q_{1}$, $q_{2}$ represents pixel coordinates of two images; *F* represents the Fundamental Matrix.

In VPR tasks, query image and database image can be respectively regarded as P1, P2 in Figure 6. According to the extracted keypoint descriptors, the Fundamental Matrix will be computed; meanwhile, the Ransac algorithm will distinguish the principle followed “inliners” that follow the principle and the “outliners” where the principle is unsuitable. The more “inliners” there are, the more similar the query image and database image are. Figure 5 visualizes the comparison among ORB, SIFT and Geodesc descriptor matching on a pair of images with slightly different perspectives. It can be seen that, compared to traditional SIFT descriptors, the matching has greatly improved with geodesc descriptors in terms of both the well-matched points and mis-matched captions.

In our panoramic localizer, the panoramic plane images include four ordinary forward-facing images from four directions. Under such conditions, the corresponding parts of query images and database images will be computed to have the “inliners” respectively and the sum of “inliners” can determine the best locations.

## 4. Experiments

In this section, local forward-facing images from each location of Pittsburgh dataset [54] will first be stitched to panoramic surround images, which serve as training and validation data of NetVLAD. Subsequently, the evaluation on the performance of only the coarse stage will be elaborated. Finally, the comparison experiments between coarse stage and coarse-to-fine processing, as well as the assessment of the panoramic localizer will be conducted. All the experiments are temporarily calculated on the PC (CPU: Intel(R) Xeon(R) CPU E5-2630 v4, Memory: 64G, GPU: Nvidia GTX1080 Ti), but deployment on mobile or wearable devices will be attempted in the future.

### 4.1. Pittsburgh Dataset Stitching

Image diversity and variety are imperative demands for NetVLAD training data. It is found that the Pittsburgh dataset with 254,064 perspective images, generated from 10,586 Google Street View locations of the Pittsburgh area, can perceive more scenarios and omnidirectional perception. There are two yaw directions and 12 pitch directions ([0, 30,…, 360]). A total of 24 images in one location of Pittsburgh dataset is associated with the same GPS location. For the requirements on horizontal training and validation data, and out of the expectation to save memory and promote computation efficiency, we only stitch the lower 12 perspective images into panoramic surround images [55], which are believed to cover the perfect view for pedestrian and vehicle navigation. The upper images are discarded due to the large areas of sky views they contain, which are regarded as useless for VPR tasks. Figure 7a shows random lower 12 perspective images in one location from the Pittsburgh dataset, while Figure 7b shows the stitched result of the 12 images. The black blurs existed on the edge of the panoramic surround images, which is reasonable after stitching. However, there is a low probability of failing in the stitching procedure, when the features between adjacent images are too similar or too few. Figure 7c shows an example of the mis-stitched results, which are thrown away and not included in the training. We transform images from the Pitt250k train subset and Pitt250k val subset to the stitched panoramic surround images according to the image sequence numbers. Finally, 3632 database and 296 query panoramic surround images are contained in the Pitt250k train subset, while 3207 database and 294 query panoramic surround images are included in the Pitt250 val subset. We train NetVLAD network on the Pitts250k train subset.

### 4.2. Validation on Coarse Stage: Rough Candidates Can Be Selected

Experimental verification of merely coarse stage is essential. One critical point is to validate whether the coarse stage can select the top-K candidates at a high precision. If merely coarse stage performs excellently, the whole system is surely successful and robust. To carry out the panorama-to-panorama matching, we feed experiment data in panoramic surround form into the well trained backbone+NetVLAD network (backbones include AlexNet, VGG16, ResNet18, and MobileNet V2).

Firstly, we verify the feasibility of the NetVLAD models on a stitched Pitt250k val subset. Recall@N denotes the recall value, where only if one of the top-N results is in the range of ground truth is it regarded as an accurate localization. Matching results shown in Table 2 indicate that the models yield a remarkable performance on not only rough range, but even on fine range. The best performed model ResNet18+NetVLAD on rough range reaches a high recall@20 of 0.9932 and the finest top-1 result gets to 0.9456. Even the top 12 images and mis-stitched images are discarded, which shows a great success of panorama-to-panorama NetVLAD models. Nevertheless, train subset and val subset are from the same Pittsburgh dataset with similar illumination and perspective, while the versatility and generalization capacity of the models should also be taken into consideration by further verifying the performances in previously unseen domains.

Secondly, the generalization ability demonstration, as a more challenging task, is indispensable. Thereby, we validate our coarse stage on the summer night subset from MOLP dataset [56] with totally different illumination and environment, across day-night cycling and reverse traversing directions: forward traversal as database, and backward traversal as query, where the tolerance is set to five images before and after the ground truth. The results shown in Table 3 further validate the generalization capability for large-scale images and discriminability for appearance changes. The best recall@20 reaches 0.8657 with a VGG16+NetVLAD model which proves that the rough range can be relatively accurate to contain correct locations. Our panorama-to-panorama NetVLAD models even learn to be irrelevant with day-night appearances, although the training data have no night information.

Thirdly, assessing the performance in our real-world campus scenario should also be convincing to further verify the generalization capacity. The panoramic Yuquan dataset [2], which was previously collected with PAL on a three-kilometer route in Zhejiang University, can perfectly reflect a real-world localization task. Subset Afternoon1 and subset Afternoon2 were both captured on sunny afternoons but from different traverses and another subset Dusk was captured at dusk. Setting the Afternoon1 subset as a database and an Afternoon2 subset as query, respectively, ground truth is annotated based on GPS information and the tolerance distance between two images is set to 50 m. Localization results of coarse stage on the Yuquan dataset are shown in Table 4 across different traversals and shown in Table 5 when confronting illumination variations. The best performed VGG16+NetVLAD model shows a high level of location accuracy, which reaches 0.9589 at recall@20 across different traversals in the afternoon, and achieves 0.9057 at recall@20 across afternoon and dusk variation. Even with the top-1 recall rate, our method surpasses the Panoramic Annular Localizer proposed by Cheng et al. [2] by large margins. Under the same conditions, our proposal achieves 0.8173 across Afternoon1 and Afternoon2, and 0.5893 confronting Afternoon1 and Dusk, clearly outstripping their 0.4524 and 0.3289 [2].

Three sets of experiments on “backbone with NetVLAD” models across totally different environments, illumination, and traversals demonstrate that our models get excellent performance on choosing the top-K rough candidates, and even maintain precision on the fine results that is not bad. Among these “backbone with NetVLAD” models, “VGG16 with NetVLAD” and “ResNet18 with NetVLAD” model achieve the highest accuracy, but, because the model generalization ability differs in disparate datasets, the best performer often has little fluctuation. “MobileNet V2 with NetVLAD” model shows slightly lower accuracy, for its precision is balanced while ensuring real-time performance, but still maintains a high level, especially in the coarse results. In a word, the coarse stage makes superior contributions on determining the rough candidates and narrowing down the calculation range of the following fine stage.

### 4.3. The Coarse-to-Fine Validation

The Chengyuan dataset is collected by a fully electric instrumented vehicle on Chengyuan campus in Gongshu District in Hangzhou, China at a speed of 12–15 km/h, and recorded in ten seconds per frame, to facilitate the study of our panoramic localizer for intelligent vehicles applications. The fully electric instrumented vehicle is integrated with a GPS tracking locator, 3 LiDARs, a PAL system, and a stereo camera, which can be seen as a data collection platform. The trajectory of the navigation route and the fully electric instrumented vehicle are shown in Figure 8. We only adopt the PAL to conduct the panoramic localizer research, but multiple works and studies will be carried out utilizing the fully electric instrumented vehicle in the near future. The setting of the acquisition program enables the real-time output of the two forms of unfolded images shown in Figure 4c [39], and the requirements of coarse stage and fine stage can be met simultaneously. This dataset covers the variations of summer scenarios in subset 2 and winter scenarios in subset 1, as well as sunny afternoon scenarios in subset 1 and cloudy morning scenarios in subset 3, where subset 1 is set as the database and the left two subsets are set as query, respectively.

In the coarse stage, NetVLAD descriptors will be obtained by the above method, which will be utilized to determine top-K candidates. Only the top-10 database images selected by the coarse stage need to conduct the following fine stage. In the fine stage, four plane images will be unfolded by the second processing method of Figure 4c from one panoramic annular image. The corresponding part of the four plane images between database and query images will be used to extract Geodesc descriptors—the most robust keypoint descriptors to compute Fundamental Matrix as a mapping relationship, respectively. Inliners and outliners will be distinguished with a Fundamental Matrix, and the total inliner number of the four parts will help to determine the final top1 result, or in other words, finer result. Table 6 and Table 7 give the localization results of coarse-to-fine system, and Figure 9 draws line charts of the matching results. The coarse matching results are drawn as a dashed line, while the results of coarse-to-fine matching are given with solid lines. Specifically, we evaluate in two settings, by considering 1 or 3 images before and after the ground truth as accurate localization. As shown in Figure 9, the challenges brought by season changes seem to be more serious than those from weather changes, for the recall in Figure 9a is not as high as Figure 9b. We can also see that, after a combination of coarse matching and fine matching, the finer results improve dramatically compared to only through the coarse matching. In this sense, our coarse-to-fine localizer obtains a great success verified by both numerical and qualitative results.

### 4.4. Efforts on Real-Time Locating

Response time and computation-efficient localization are of great concern for assisted navigation to satisfy high-level applications. Nevertheless, our coarse-to-fine localizer goes through two rounds of sophisticated stages, and adopts time-consuming CNN-based NetVLAD feature extractors. Moreover, we utilize panoramas that usually have high resolution for training and testing, which entails inefficient computation and time cost if the mobile computing platform lacks sufficient resources and computing power. One target is to search for a lightweight CNN network like MobileNet V2, as we analyze in 3.2.1 and show detailed experiments above, which can reduce the computation resource to 1/K×K and maintain a certain accuracy. We collect average consumption time per image on four backbones based NetVLAD models of coarse stage on the Chengyuan dataset, whose performances respectively show as: 0.0594 s on AlexNet, 0.3379 s on VGG16, 0.1951 s on ResNet18 and 0.1097 s on MobileNet V2. AlexNet benefits from its simple structure and shallow network, and thereby achieves the least consuming time while causes a great loss of accuracy in the meantime. MobileNet V2 does keep time-efficiency as it is analyzed and keeps a comparatively high accuracy, compared to VGG16 and the ResNet18 based model, although it is not so accurate as the top-performing standard convolutions based CNNs, but sufficient to cover our demands.

However, the computation-expensive deep-learning based algorithm Geodesc, which processes from keypoint detection, is a very time-consuming operation to crop these keypoints as 32 × 32 sized 32 squares. The computationally-intensive CNN-based feature extraction does provide high-level and ideal localization results, but real-time prediction cannot be satisfied. For this reason, we also leverage faster descriptors such as SIFT and ORB, and compare with Geodesc in Table 8 and Table 9. Only the condition with one image before and after the ground truth is considered as accurate localization.

Another point that should be mentioned is that usually there are many meaningless and relatively plain scenes in the images, like ground and sky, which are insignificant to VPR. Thus, we implement BING [57]—a salient region detection algorithm, on the top-k candidates selected by the coarse stage, so that critical point detection in non-critical areas is avoided, to save time and computation. The pipeline of deciding a top-1 candidate after implantation of BING salient region detection into coarse-to-fine localizer is shown in Figure 10. The results are also given in Table 8 and Table 9. We also draw line charts of matching results on Chengyuan dataset with simplified models in Figure 11.

Time consumptions in the fine stage with ORB, SIFT, Geodesc and BING+SIFT, respectively, on the Chengyuan dataset are also computed and recorded in Table 10. Because the time cost of the fine stage depends largely on the rough candidate number K selected by of the coarse stage, the values listed in Table 10 represent the consumed time of a matching pair. By the way, the BING+SIFT time measurement is to record time after discarding cropping time of BING. This operation is so time-consuming that adding this part of time to the result will be a departure from our original intention to calculate only the characteristics of critical areas. Alternative time-saving solutions which can replace cropping will be attempted in the future.

By analyzing the data and line charts above, we can recognize that the time consumption of the coarse and fine stages has a great disparity, relatively good real-time rows can be achieved in the coarse stage, while once the selected candidate number K is large, the time cost of matching per image will be K times the value above listed in Table 10. It seems that only ORB matching in fine stage can have a relatively good real-time performance, but it performs decently in the localization accuracy. In this regard, on the occasions where real-time processing must be guaranteed, only utilizing the coarse stage or adopting “Coarse stage+ORB” scheme are advisable solutions. The “Coarse stage+Geodesc” combination brings an explosion in time consumption due to the time complexity of Geodesc, although it possesses the highest accuracy superiority in maintaining the accuracy of positioning. The “BING” operation can slightly improve the time saving performance but can also keep a relatively high accuracy, and the SIFT descriptors show certain robustness in localization accuracy. The “MobileNet V2 with NetVLAD+BING+SIFT” model seems to be a good compromise between response time and precision, being a critical player that strikes an excellent balance between efficiency and accuracy. Figure 12 displays some correct visual localization results of the proposed panoramic localizer on the Chengyuan dataset.

## 5. Conclusions

In this work, we have proposed a conceptually simple coarse-to-fine descriptors based panoramic localizer for navigation assistance to resist severe challenges across different traversals, weather variations, day-night cycling, illumination changes, and viewpoint transformations in visual place recognition (VPR) tasks. The panoramic localizer utilizes panoramas with large Field of View (FoV) to perceive omnidirectional environments and yields the accurate localization via a carefully designed coarse-to-fine framework. In the coarse stage, several CNN backbones are respectively integrated in a NetVLAD layer with specific learning capacity, constructing a global panorama-to-panorama feature detector to determine the rough top-K candidates. The fine stage provides an intelligent keypoint descriptor catcher, especially when employing the deep learning based and high-accuracy Geodesc keypoint descriptors, in order to further locate the finest positions.

Comprehensive experiments on the coarse stage and the coarse-to-fine process have shown excellent performances of the panoramic localizer, especially the dramatic promotion of accuracy after contributions of the fine stage, and the “VGG16 with NetVLAD+Geodesc” model maintains high-level results on the Chengyuan dataset at recall@1, which confirms the stability and perfect generality of our panoramic localizer. Furthermore, to guarantee the real-time performance, lightweight MobileNet V2 based NetVLAD, the salient region based algorithm BING and faster handcrafted keypoints SIFT, ORB are attempted on our system, which achieve relatively high accuracy, and results in less time consumption in the meantime. In summary, the coarse stage possesses an excellent ability to pick out rough candidates and even the finer range can maintain a high-level precision, while the fine stage performs a great leap in accuracy, which can reach nearly perfect localization results. In addition, the real-time considerations also prove to be effective and superior. Our panoramic localizer greatly outperforms state-of-the-art VPR works, achieving high localization results and practical application possibility. In the future, we aim to transplant our panoramic localizer to a mobile navigation device and further promote the localization ability, to provide convenience for pedestrian navigation.

## Figures and Tables

**Figure 1 sensors-20-04177-f001:**
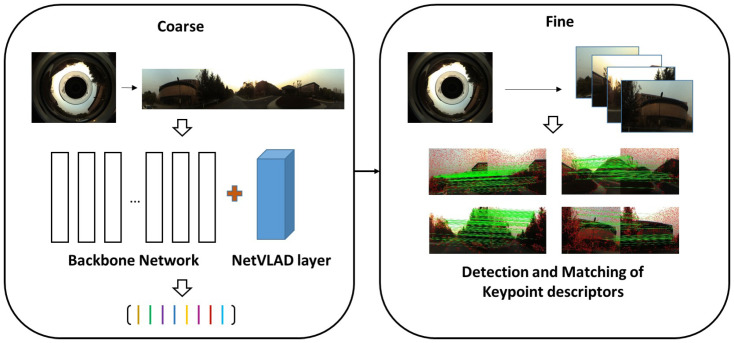
Overview of the proposed coarse-to-fine localizer. In the localization phase, NetVLAD descriptors and keypoint descriptors will be detected in the coarse stage and fine stage, respectively. After that, Brute Force (BF) matching and Fundamental Matrix mapping will be conducted, respectively.

**Figure 2 sensors-20-04177-f002:**
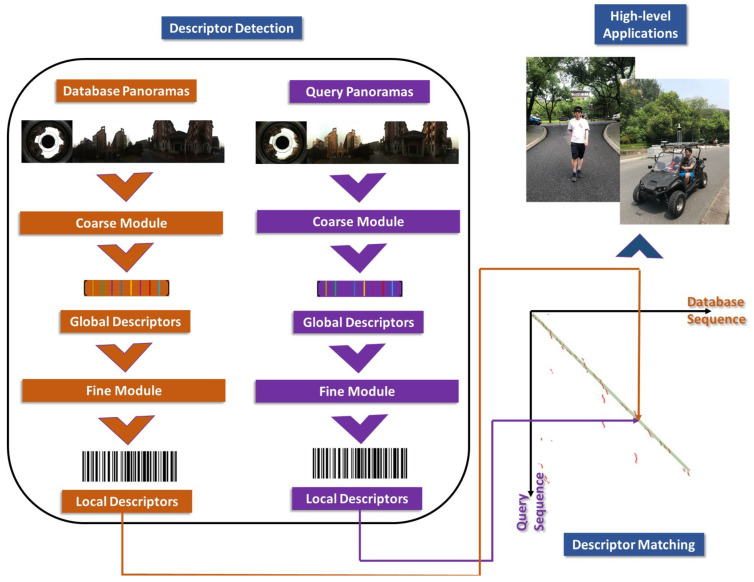
Panoramic descriptor detection, descriptor matching procedures in VPR, and their high-level navigation assistance applications for VIP or autonomously driving vehicles. (The curve at the right bottom represents the descriptor matching relationship between query images and the corresponding best matched database images. The red points refer the matching instance points and the green line is on behalf of the ground truth.)

**Figure 3 sensors-20-04177-f003:**
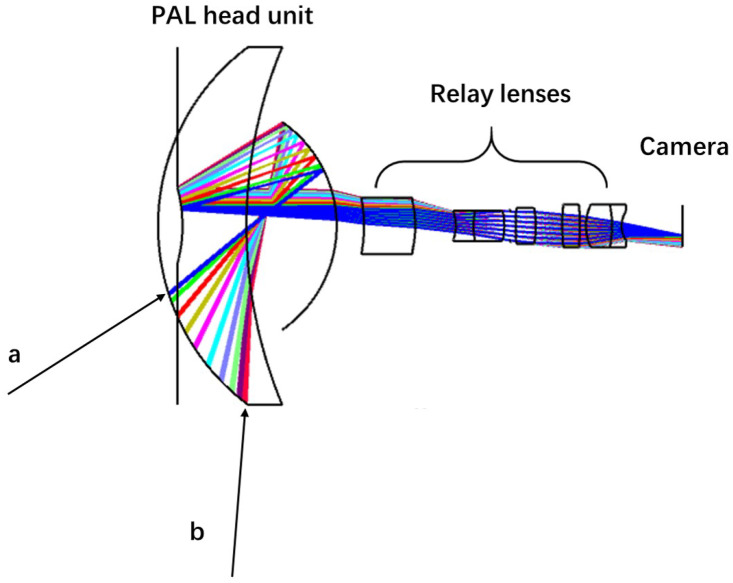
Basic structure and the imaging optical path of PAL (a and b represent the maximum incident angle range of PAL).

**Figure 4 sensors-20-04177-f004:**
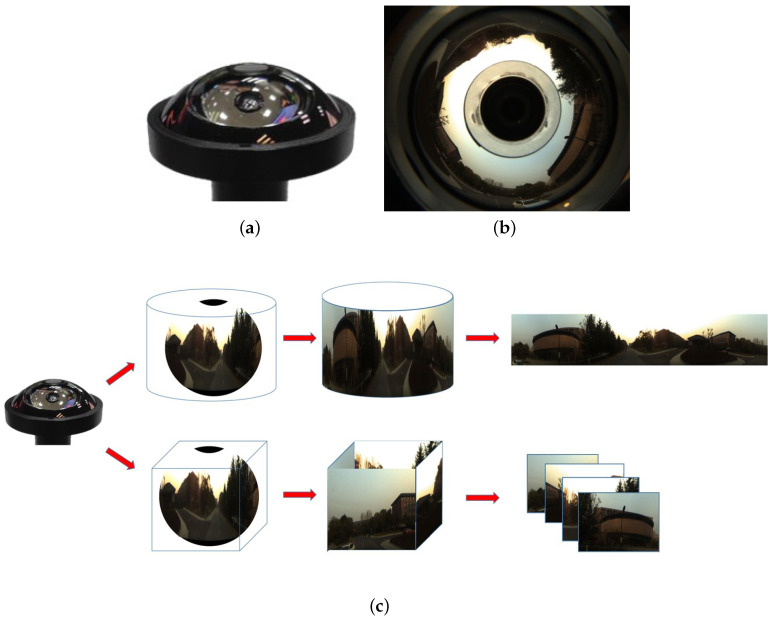
(**a**) PAL; (**b**) a panoramic annular image imaged by PAL; (**c**) two panoramic annular image processing forms. The upper method projects the unit spherical surface onto a cylinder, while the bottom method projects the unit spherical surface onto a cube.

**Figure 5 sensors-20-04177-f005:**
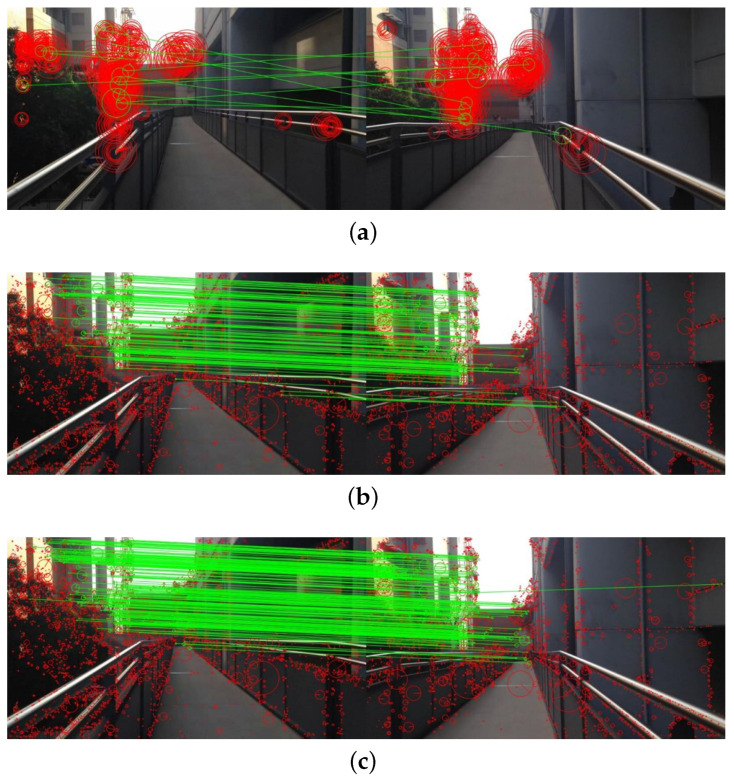
Comparison of matching results among (**a**) ORB descriptors, (**b**) SIFT descriptors, and (**c**) Geodesc descriptors.

**Figure 6 sensors-20-04177-f006:**
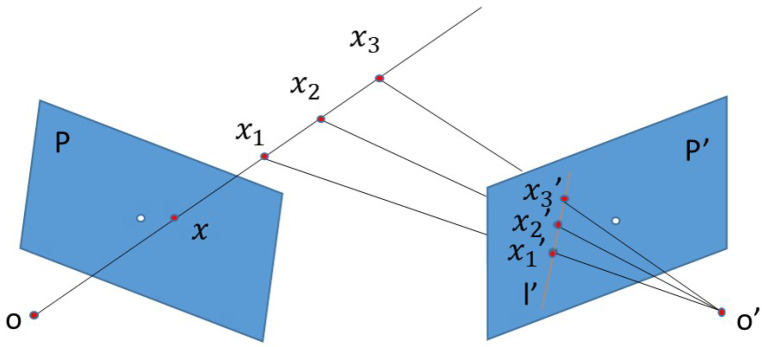
Matching principle of the Fundamental Matrix for keypoints.

**Figure 7 sensors-20-04177-f007:**
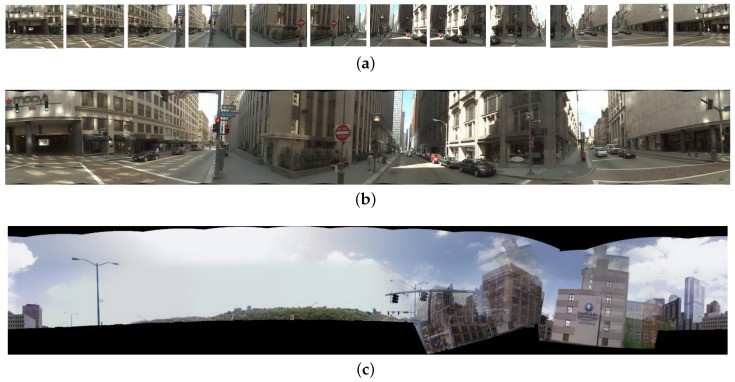
(**a**) random bottom 12 images in one location from Pittsburgh dataset; (**b**) the stitched panorama of (**a**); (**c**) a mis-stitched result.

**Figure 8 sensors-20-04177-f008:**
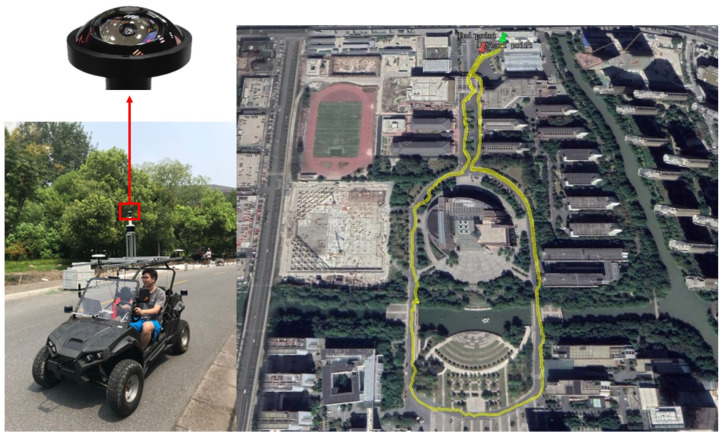
The trajectory of the navigation route and the fully electric instrumented vehicle.

**Figure 9 sensors-20-04177-f009:**
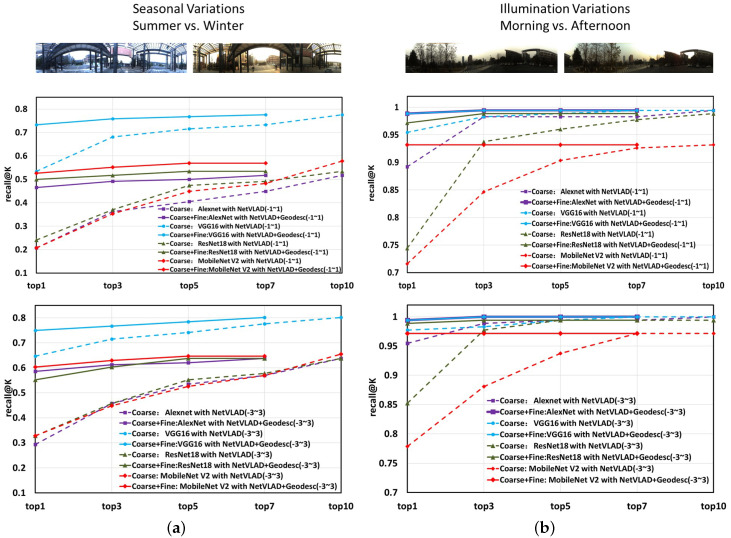
Line chart of matching results on Chengyuan dataset. (**a**) subset 2 as query and subset 1 as database; (**b**) subset 3 as query and subset 1 as query. (Coarse means the coarse stage, Coarse+Fine means combination of the coarse stage and the fine stage. −1–1 means 1 image before and after the ground truth are setting as accurate localization, −3–3 means 3 images before and after the ground truth are setting as accurate localization.)

**Figure 10 sensors-20-04177-f010:**
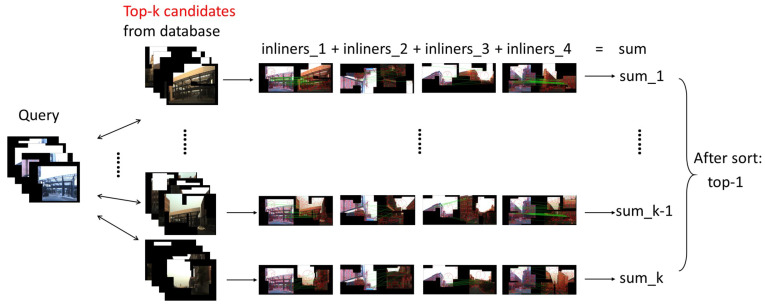
Pipeline of deciding top-1 candidate after implantation of BING salient region detection into a coarse-to-fine localizer.

**Figure 11 sensors-20-04177-f011:**
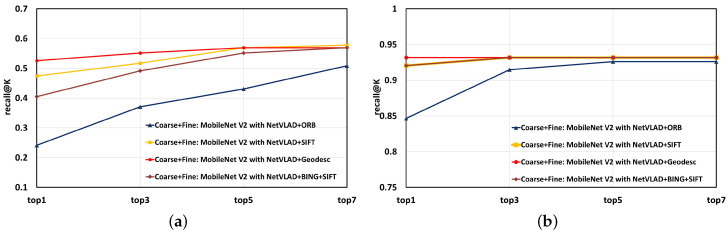
Line chart of matching results when simplify models on the Chengyuan dataset. (**a**) subset 2 as query and subset 1 as database; (**b**) subset 3 as query and subset 1 as query. (Coarse means the coarse stage, Coarse+Fine means combination of the coarse stage and the fine stage.)

**Figure 12 sensors-20-04177-f012:**
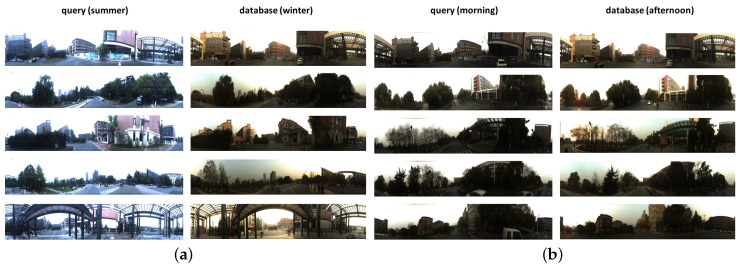
The visual localization results. (**a**) subset 2 as query and subset 1 as database; (**b**) subset 3 as query and subset 1 as database. The left column contains the query images, and the right column presents the corresponding database images.

**Table 1 sensors-20-04177-t001:** The architecture of four backbones: AlexNet, VGG16, ResNet18, MobileNet V2.

Backbone	Architecture
AlexNet	
VGG16	
ResNet18	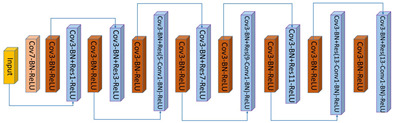
MobileNet V2	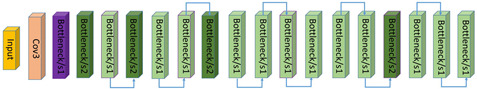

**Table 2 sensors-20-04177-t002:** Localization results of coarse stage on the Pitt250 val subset.

Model	Recall@1	Recall@5	Recall@10	Recall@20
Coarse: AlexNet with NetVLAD	0.7585	0.9354	0.9226	0.9762
Coarse: VGG16 with NetVLAD	0.9286	0.9864	0.9898	0.9898
**Coarse: ResNet18 with NetVLAD**	**0.9456**	**0.9864**	**0.9932**	**0.9932**
Coarse: MobileNet V2 with NetVLAD	0.7551	0.9626	0.9864	0.9932

**Table 3 sensors-20-04177-t003:** Localization results of coarse stage on the summer night subset of MOLP dataset [56]–Backward as query, Forward as database.

Model	Recall@1	Recall@5	Recall@10	Recall@20
Coarse: AlexNet with NetVLAD	0.2525	0.5170	0.6453	0.7976
**Coarse: VGG16 with NetVLAD**	**0.4008**	**0.6613**	**0.7495**	**0.8417**
**Coarse: ResNet18 with NetVLAD**	**0.3006**	**0.6253**	**0.7635**	**0.8657**
Coarse: MobileNet V2 with NetVLAD	0.2325	0.5090	0.6553	0.7675

**Table 4 sensors-20-04177-t004:** Localization results of coarse stage on the Yuquan dataset [2]–Afternoon2 as query, Afternoon1 as database.

Model	Recall@1	Recall@5	Recall@10	Recall@20
Coarse: AlexNet with NetVLAD	0.7323	0.8470	0.8966	0.9334
**Coarse: VGG16 with NetVLAD**	**0.8173**	**0.8924**	**0.9292**	**0.9589**
Coarse: ResNet18 with NetVLAD	0.6388	0.7975	0.8343	0.8697
Coarse: MobileNet V2 with NetVLAD	0.6969	0.8144	0.8569	0.9178
Panoramic Annular Localizer	0.4524			

**Table 5 sensors-20-04177-t005:** Localization results of coarse stage on the Yuquan dataset [2]–Dusk as query, and Afternoon1 as database.

Model	Recall@1	Recall@5	Recall@10	Recall@20
Coarse: AlexNet+NetVLAD	0.3788	0.5181	0.6313	0.7605
**Coarse: VGG16+NetVLAD**	**0.5893**	**0.8200**	**0.7057**	**0.9057**
Coarse: ResNet18+NetVLAD	0.3556	0.6038	0.7068	0.7881
Coarse: MobileNet V2+NetVLAD	0.3382	0.5631	0.6560	0.7591
Panoramic Annular Localizer	0.3289			

**Table 6 sensors-20-04177-t006:** Localization results of the coarse-to-fine system on Chengyuan dataset across summer and winter—subset 2 as query and subset 1 as database.

Model	Recall@1	Recall@3	Recall@5	Recall@7	Recall@10
Coarse: AlexNet with NetVLAD(-1-1)	0.2069	0.3621	0.4052	0.4483	0.5172
**Coarse-to-fine: AlexNet with NetVLAD+Geodesc(-1-1)**	**0.4655**	**0.4914**	**0.5000**	**0.5172**
Coarse: VGG16 with NetVLAD(-1-1)	0.5345	0.6810	0.7155	0.7328	0.7759
**Coarse-to-fine: VGG16 with NetVLAD+Geodesc(-1-1)**	**0.7328**	**0.7586**	**0.7672**	**0.7759**
Coarse: ResNet18 with NetVLAD(-1 1)	0.2414	0.3707	0.4741	0.4914	0.5345
**Coarse-to-fine: ResNet18 with NetVLAD+Geodesc(-1-1)**	**0.5000**	**0.5172**	**0.5345**	**0.5345**
Coarse: MobileNet V2 with NetVLAD(-1-1)	0.2068	0.3534	0.4483	0.4828	0.5776
**Coarse-to-fine: MobileNet V2 with NetVLAD+Geodesc(-1-1)**	**0.5259**	**0.5517**	**0.5690**	**0.5690**
Coarse: AlexNet with NetVLAD(-3-3)	0.2931	0.4569	0.5345	0.5690	0.6379
**Coarse-to-fine: AlexNet with NetVLAD+Geodesc(-3-3)**	**0.5862**	**0.6121**	**0.6207**	**0.6379**
Coarse: VGG16 with NetVLAD(-3-3)	0.6466	0.7155	0.7414	0.7759	0.8017
**Coarse-to-fine: VGG16 with NetVLAD+Geodesc(-3-3)**	**0.7500**	**0.7672**	**0.7845**	**0.8017**
Coarse: ResNet18 with NetVLAD(-3-3)	0.32760	0.4570	0.5517	0.57760	0.6379
**Coarse-to-fine: ResNet18 with NetVLAD+Geodesc(-3-3)**	**0.5517**	**0.6035**	**0.6379**	**0.6379**
Coarse: MobileNet V2 with NetVLAD(-3-3)	0.3276	0.4483	0.5259	0.5690	0.6552
**Coarse-to-fine: MobileNet V2 with NetVLAD+Geodesc(-3-3)**	**0.6034**	**0.6293**	**0.6466**	**0.6466**	

**Table 7 sensors-20-04177-t007:** Localization results of a coarse-to-fine system on the Chengyuan dataset across summer and winter—subset 2 as query, subset 1 as database.

Model	Recall@1	Recall@3	Recall@5	Recall@7	Recall@10
Coarse: AlexNet with NetVLAD(-1-1)	0.8921	0.9830	0.9830	0.9830	0.9943
**Coarse-to-fine: AlexNet with NetVLAD+Geodesc(-1-1)**	**0.9886**	**0.9943**	**0.9943**	**0.9943**
Coarse: VGG16 with NetVLAD(-1-1)	0.9546	0.9830	0.9886	0.9943	0.9943
**Coarse-to-fine: VGG16 with NetVLAD+Geodesc(-1-1)**	**0.9886**	**0.9943**	**0.9943**	**0.9943**
Coarse: ResNet18 with NetVLAD(-1 1)	0.7443	0.9375	0.9602	0.9773	0.9886
**Coarse-to-fine: ResNet18 with NetVLAD+Geodesc(-1-1)**	**0.9716**	**0.9886**	**0.9886**	**0.9886**
Coarse: MobileNet V2 with NetVLAD(-1-1)	0.7159	0.8466	0.9034	0.9261	0.9318
**Coarse-to-fine: MobileNet V2 with NetVLAD+Geodesc(-1-1)**	**0.9318**	**0.9318**	**0.9318**	**0.9318**
Coarse: AlexNet with NetVLAD(-3-3)	0.9546	0.9886	0.9943	0.9943	1.0000
**Coarse-to-fine: AlexNet with NetVLAD+Geodesc(-3-3)**	**0.9943**	**1.0000**	**1.0000**	**1.0000**
Coarse: VGG16 with NetVLAD(-3-3)	0.9773	0.9830	0.9943	1.0000	1.0000
**Coarse-to-fine: VGG16 with NetVLAD+Geodesc(-3-3)**	**0.9943**	**1.0000**	**1.0000**	**1.0000**
Coarse: ResNet18 with NetVLAD(-3-3)	0.8523	0.9773	0.9943	0.9943	0.9943
**Coarse-to-fine: ResNet18 with NetVLAD+Geodesc(-3-3)**	**0.9886**	**0.9943**	**0.9943**	**0.9943**
Coarse: MobileNet V2 with NetVLAD(-3-3)	0.7784	0.8807	0.9375	0.9716	0.9716
**Coarse-to-fine: MobileNet V2 with NetVLAD(-3-3)**	**0.9716**	**0.9716**	**0.9716**	**0.9716**	

**Table 8 sensors-20-04177-t008:** Localization results of coarse-to-fine system when simplifying models on the Chengyuan dataset across summer and winter—subset 2 as query, subset 1 as database.

Model	Recall@1	Recall@3	Recall@5	Recall@7
Coarse+Fine: MobileNet V2 with NetVLAD+ORB	0.2414	0.3707	0.4310	0.5086
Coarse+Fine: MobileNet V2 with NetVLAD+SIFT	0.4741	0.5172	0.5690	0.5776
**Coarse+Fine: MobileNet V2 with NetVLAD+Geodesc**	**0.5259**	**0.5517**	**0.5690**	**0.5690**
Coarse+Fine: MobileNet V2 with NetVLAD+BING+SIFT	0.4052	0.4914	0.5517	0.5690

**Table 9 sensors-20-04177-t009:** Localization results of coarse-to-fine system when simplifying models on the Chengyuan dataset across morning and afternoon—subset 3 as query, subset 1 as database.

Model	Recall@1	Recall@3	Recall@5	Recall@7
Coarse+Fine: MobileNet V2 with NetVLAD+ORB	0.8466	0.9148	0.9261	0.9261
Coarse+Fine: MobileNet V2 with NetVLAD+SIFT	0.9205	0.9318	0.9318	0.9318
**Coarse+Fine: MobileNet V2 with NetVLAD+Geodesc**	**0.9318**	**0.9318**	**0.9318**	**0.9318**
Coarse+Fine: MobileNet V2 with NetVLAD+BING+SIFT	0.9205	0.9318	0.9318	0.9318

**Table 10 sensors-20-04177-t010:** The consumed time comparison of per matching pair of the fine stage with ORB, SIFT, Geodesc, and BING+SIFT, respectively, on the Chengyuan dataset.

Model	ORB	SIFT	Geodesc	SIFT+BING
time	0.1490 s	1.5307 s	11.3295 s	0.3657 s

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
