# Peer review of "A Panoramic Localizer Based on Coarse-to-Fine Descriptors for Navigation Assistance"

_sensors, 2020, doi:10.3390/s20154177_

Round 1

Reviewer 1 Report

This article describes the 2-step method used to first select the best candidates using a coarse matching method and subsequently perform the finer matching using more computationally methods. The concept makes perfect sense, although it is not really novel. For applications that require VPR, panoramic annular images are argued to be more suitable, which also makes sense, although not completely novel. The interesting aspect here is typically the experimental analysis using images collected in different environments.

Some details are missing, which could be included for a more complete analysis. Some writing styles could also be improved.

  1. L25-28 analyzes the downside of GNSS, but the analysis is insufficient. There are many more localization techniques used in GNSS-denied areas that are not considered : WIFI, Pseudo-satellite, UWB ranging, sensor fusion (INS+GNSS+Camera).
  2. Figure 1 seems to be stating the obvious: The curve comparing descriptors of query panoramas and database panoramas? It is a little bothering to see a huge figure stating the obvious.
  3. Section 2 on related work: Reference(s) for PAL should be cited.
  4. Section 3.3.2 Keypoint matching with fundamental matrix and Ransac algorithm is basic. It can be “recapitulated” here, but the reference should also be cited.
  5. L234-246: Citation missing: which programs or publication of methods used to obtain plane images from panoramic annular images?
  6. L426: What are the characteristics of the fully electric instrumented vehicle? The speed of the vehicle should be provided to know how the images of the dataset are acquired.
  7. Section 4.4: This section should provide results on the real-time capacity. It is claimed that response time can be reduced, but there is no analysis or comparisons made on the response time. Neither is there an order or magnitude given.
  8. The authors also did not provide information on the kind of platform the experiments were performed. Is it a desktop PC, a mobile device or a wearable? What kind of computational performance are we referring to?

There are quite a number of typos and mistakes in the article. Here are a few listed. But the authors may want to proofread the article again.

  • L3: existing positions
  • L58: panoramas minimize
  • L82: give results
  • L87: taken into consideration
  • Figure 2: “Keypoint descriptors”            
  • L150: have been utilized
  • L193: Cite author name properly: Iscen et al.
  • L254: should discard
  • L330 are
  • Figure 11: Labelling of the subfigures is missing
  • L364: “law” images?
  • L472: Table x?

Author Response

Response to Reviewer 1 Comments

Point 1: This article describes the 2-step method used to first select the best candidates using a coarse matching method and subsequently perform the finer matching using more computationally methods. The concept makes perfect sense, although it is not really novel. For applications that require VPR, panoramic annular images are argued to be more suitable, which also makes sense, although not completely novel. The interesting aspect here is typically the experimental analysis using images collected in different environments. Some details are missing, which could be included for a more complete analysis. Some writing styles could also be improved.

Response 1: Thank you very much for your valuable comments. We are very pleased that the reviewer affirmed the contribution of the article. The authors are grateful to the reviewer for the valuable suggestions. As for the missing details, we next conduct a more complete analysis according to the reviewer’s comments.

Point 2: L25-28 analyzes the downside of GNSS, but the analysis is insufficient. There are many more localization techniques used in GNSS-denied areas that are not considered: WIFI, Pseudo-satellite, UWB ranging, sensor fusion (INS+GNSS+Camera).

Response 2: Thank you very much for your valuable comments. In order to address this comment, we have added more detailed analysis among GNSS, WIFI, Pseudo-satellite, UWB ranging, sensor fusion (INS+GNSS+Camera), to highlight advantages and characteristics of VPR, and cited some related references in Section 1. Introduction, paragraph one and paragraph two. Please refer to the revised version.

Point 3: Figure 1 seems to be stating the obvious: The curve comparing descriptors of query panoramas and database panoramas? It is a little bothering to see a huge figure stating the obvious.

Response 3: Thank you very much for your valuable comments. Figure 1 has been modified and the descriptor detection part has been unfolded to describe the procedure of the coarse module and fine module. The curve represents the descriptor matching relationship between query images and the corresponding best matched database images, the red points refer the matching instance points and the green line is on behalf of the ground truth, if most of the red points can be covered by the green line, the matching results can be convincing. More detailed legend has been given.

Point 4: Section 2 on related work: Reference(s) for PAL should be cited.

Response 4: Thank you very much for your valuable comments. Three references as follow for PAL have been cited in Section 2.2. We have also modified the imaging principle of PAL (Figure 3), and make plenty of improvement and give more detailed introduction in Section 2.1. Please refer to the revised version.

[34] Zhou, X.; Bai, J.; Wang, C.; Hou, X.; Wang, K. Comparison of two panoramic front unit arrangements in design of a super wide angle panoramic annular lens. Applied Optics2016,55, 3219–3225.

[35] Sun, D.; Huang, X.; Yang, K.  A multimodal vision sensor for autonomous driving.  Counterterrorism, Crime Fighting, Forensics, and Surveillance Technologies III. International Society for Optics and Photonics, 2019, Vol. 11166, p. 111660L.

[36] W. Hu, et al. An indoor positioning framework based on panoramic visual odometry for visually impaired people. Measurement Science and Technology, 2019.

Point 5: Section 3.3.2 Keypoint matching with fundamental matrix and Ransac algorithm is basic. It can be “recapitulated” here, but the reference should also be cited.

Response 5: Thank you very much for your valuable comments. To address this comment, the corresponding reference has been cited in Section 3.3.2.[51]

[51] Zhang, Z.; Loop, C. Estimating the fundamental matrix by transforming image points in projective space. Computer vision and image understanding2001,82, 174–180.

Point 6: L234-246: Citation missing: which programs or publication of methods used to obtain plane images from panoramic annular images?

Response 6: Thank you very much for your valuable comments. Hu et al. [36] proposed a multiple-pinhole rectification of raw PAL images, which allows the undistorted images after multiple-pinhole rectification satisfy the perspective projection rules, by mapping a PAL image onto a unit sphere, extending the unit sphere to a cube and then using four virtual pinhole cameras to capture the side surfaces of the cube. This multiple-pinhole rectification method has been referred and utilized in our panoramic localizer to generate plane images from panoramic annular images. Related reference has been attached and more detailed descriptions have been given in Section 2.1 and Section 3.1. Please refer to the revised version

[36] W. Hu, et al. An indoor positioning framework based on panoramic visual odometry for visually impaired people. Measurement Science and Technology, 2019.

Point 7: L234-246:  What are the characteristics of the fully electric instrumented vehicle? The speed of the vehicle should be provided to know how the images of the dataset are acquired.

Response 7: Thank you very much for your valuable comments. The fully electric instrumented vehicle integrates a GPS tracking locator, 3 LiDAR sensors, a PAL system and a stereo camera, which can be seen as a data collection platform. We only adopt the PAL to conduct the panoramic localizer research, but multiple works and studies will be carried out by utilizing the fully electric instrumented vehicle in the near future. Chengyuan dataset was collected by the fully electric instrumented vehicle at a speed of 12~15 km/h, and recorded in ten seconds per frame. Characteristics about the fully electric instrumented vehicle have been added in Section 4.3. Please refer to the revised version.

Point 8: Section 4.4: This section should provide results on the real-time capacity. It is claimed that response time can be reduced, but there is no analysis or comparisons made on the response time. Neither is there an order or magnitude given.

Response 8: Thank you very much for your valuable comments. We have evaluated the coarse stage response time for average image in Section 4.4, the second half of the paragraph one and conduct a comparison and analyzation between different models. In addition, we list a Table 10 to evaluate the time consuming of per matching pair on the coarse stage, and give a deep analyse in paragraph four and paragraph five. Please refer to the revised manuscript.

Point 9: The authors also did not provide information on the kind of platform the experiments were performed. Is it a desktop PC, a mobile device or a wearable? What kind of computational performance are we referring to?

Response 9: Thank you very much for your valuable comments. Our system is temporarily calculated on the PC, but in the future, we will attempt to integrate the algorithm into mobile or wearable devices.

Point 10: There are quite a number of typos and mistakes in the article. Here are a few listed. But the authors may want to proofread the article again.

Response 10: Thank you very much for your valuable comments. We have revised the mistakes you mentioned and do a thorough examination and proofreading.

Thank you again for your constructive comments and suggestions.

Reviewer 2 Report

The authors propose a Visual Place Recognition method (VPR) based on a two coarse-to-fine steps. The source of information of the process are panoramic images from an omnidirectional visual system.

The proposal they present is very simple. In a coarse step the NetVLAD descriptors along an euclidean distance are used. In the fine step, the k selected form previous step are used to use Geodesc keypoint descriptors along with Fundamental Matrix restriction.

While the proposal is interesting, significant changes need to be made to it.

  • Many affirmations that are made are not true. There is a lot of previous work in the field of Visual Place Recognition that also uses two steps and employs panoramic images.
  • The use of panoramic images has previously been used in VPR by many authors
  • The Introduction and State of the Art sections should be completely rewritten
  • In particular, the Related Work section addresses a number of issues that are widely known. However, it does not present an adequate state of the art where works made by other authors completely related to the proposal are presented.
  • Sections 2.1, 2.2, ... are not significant to the proposal.
  • Section 2.3 is really poor and vague. A more detailed and precise analysis of other proposals in VPR in the same field with panoramic images and omnidirectional vision systems is missing
  • The system of image acquisition and conformation of the panoramic image is really confusing and there are doubts about its generality
  • It is not detailed how the origin of the cube is selected with respect to the sphere in the image acquisition system
  • The introduction and figures of the different nets used do not contribute anything significant.

Author Response

Response to Reviewer 2 Comments

Point 1: The authors propose a Visual Place Recognition method (VPR) based on a two coarse-to-fine steps. The source of information of the process are panoramic images from an omnidirectional visual system.

The proposal they present is very simple. In a coarse step the NetVLAD descriptors along an euclidean distance are used. In the fine step, the k selected form previous step are used to use Geodesc keypoint descriptors along with Fundamental Matrix restriction.

While the proposal is interesting, significant changes need to be made to it.

Response 1: Thank you very much for your valuable comments. We are very pleased that the reviewer affirmed the contributions of the article. The authors are grateful to the reviewer for the valuable suggestions. As for the problems you mentioned, we have tried our best to address.

Point 2: Many affirmations that are made are not true. There is a lot of previous work in the field of Visual Place Recognition that also uses two steps and employs panoramic images.

Response 2: Thank you very much for your valuable comments. To address this comment, we have revised all of the “sporadic researches”, or “rarely studied” on panoramic VPR, and supplemented novel and state-of-the-art “Multi-step Solutions on VPR”, and “Attempts on Panoramic VPR” in Section 2.2 and Section 2.3 respectively. Please refer to the revised version.

Point 3: The use of panoramic images has previously been used in VPR by many authors.

Response 3: Thank you very much for your valuable comments. To address this comment, we have listed several panoramic images based VPR approaches in Section 2.3, in addition, advantages and disadvantages of panoramic images in VPR are analyzed in detail in paragraph four and paragraph five of Section 1, Introduction. We have highlighted the difference of our panoramic localizer to previous works. We have introduced the proposed method by using PAL system to obtain richer panoramic image resources served for our panoramic localizer.

Point 4: The Introduction and State of the Art sections should be completely rewritten. In particular, the Related Work section addresses a number of issues that are widely known. However, it does not present an adequate state of the art where works made by other authors completely related to the proposal are presented.

Response 4: Thank you very much for your valuable comments. To address this comment, we have thoroughly modified Section 1. Introduction and Section 2. Related Work.

In Section 1, we firstly analyze and compare different localization methods, which naturally leads to the advantage of VPR, rather than only conduct the comparison between GNSS and VPR approaches.

Secondly, we analyze the challenges in VPR and list several existing VPR researches to introduce two targets we should cope with: large FoV images and two-stage based framework, so the following parts we respectively introduce and analyze these two parts in detail. Finally, we point out the contributions of our approach. In Section 2, Related Work is divided into three parts: Panoramic annular lens (PAL), Multi-step Solutions on VPR, and Attempts on Panoramic VPR, which have been completely modified. We add many state-of-the-art proposals that are related to our presented panoramic localizer. In Section 2.1, we give more detailed analysis of the PAL system. We update the image to a clearer one and cite several references. In Section 2.2, three dominating state-of-the-art multi-stage based VPR approaches are discussed. In Section 2.3, some novel proposals on panoramic VPR are introduced in detail.

Please refer to the revised manuscript.

Point5: Sections 2.1, 2.2, ... are not significant to the proposal.

Response 5: Thank you very much for your valuable comments. The previous Section 2.2 “Image Descriptors” has been deleted, however, PAL system is very significant for our panoramic localizer and a thorough introduction is necessary, which is more preferable to be set in the Related Work part, so we retain the PAL in this part but make plenty of improvements on not only figure 2 but also the content. Please refer to the revised version in Section 2.

Point 6: Section 2.3 is really poor and vague. A more detailed and precise analysis of other proposals in VPR in the same field with panoramic images and omnidirectional vision systems is missing.

Response 6: Thank you very much for your valuable comments. To improve and modify Section 2.3, we discard the behindhand method and add more novel and advanced approach, in the meanwhile, give a more detailed and precise analysis of all the listed proposals. Please refer to the revised version.

Point 7: The system of image acquisition and conformation of the panoramic image is really confusing and there are doubts about its generality.

Response 7: Thank you very much for your valuable comments. There are two processing ways of panoramas, which are utilized in our localizer system, and in regard to the more unknown panoramic plane images, we cite a reference in Section 2.1 which mentions multi-pinhole rectification in order to satisfy the perspective projection rules, by mapping a PAL image onto a unit sphere, extending the unit sphere to a cube and then using four virtual pinhole cameras to capture the side surfaces of the cube. This multiple-pinhole rectification method has been referred and utilized in our panoramic localizer to generate plane images from panoramic annular images. We have detailed introduce this part in Section 2.1 after cite the reference. In addition, in Section 3.1, we add more detailed analyse of the two processing ways of panoramas. As for the generality, we also explain in Section 3.1, if you don’t have a PAL, you can obtain the panoramic surround images by image stitching or even mobile phone scanning panoramic image function, and acquire panoramic plane images by multi-pinhole images, in order to fit our system.

Point 8: It is not detailed how the origin of the cube is selected with respect to the sphere in the image acquisition system.

Response 8: Thank you very much for your valuable comments. The origin of the cube is cited as the same places as the sphere, given by camera calibration result. We have introduced it in this Section 3.1, the paragraph two. Please refer to the revised version.

Point 9: The introduction and figures of the different nets used do not contribute anything significant.

Response 9:  Thank you very much for your valuable comments. To address this comment, we greatly foreshorten the network part of the introduction in Section 3.2.1, only the Mobilenet V2 which can reduce the number of parameters are more detailed described. In addition, we list different network architectures in Table 1, so that the differences between networks can be clearly seen.

Thank you again for your constructive comments and suggestions.

Round 2

Reviewer 2 Report

The authors have significantly improved the first version of the manuscript. They have responded adequately to most of the suggestions made in the first review.
Taking into account the modifications made, the manuscript can be accepted for publication

Author Response

Point: The authors have significantly improved the first version of the manuscript. They have responded adequately to most of the suggestions made in the first review.
Taking into account the modifications made, the manuscript can be accepted for publication

Response: Thank you very much for your valuable comments. We are very pleased that the reviewer affirmed the contribution of the article. The authors are grateful to the reviewer for the valuable suggestions. We have attached the complete manuscript in the web page.
